# Evaluation of a group antenatal care intervention in two Northern Nigerian states: Quasi-experimental study

William Douglas Evans[1]*, Jeffrey Bartlett Bingenheimer[1], Taiseer Zaman[1], Chinwe Lucia Ochu[1], Samson Babatunde Adebayo[2], Fasiku Adekunle David[2], Sani Ali Gar[2], Masduk Abdulkarim[3]

1 Milken Institute School of Public Health, The George Washington University, Washington, DC, United States of America, 2 Data Research and Mapping Consults, Abuja, Nigeria, 3 Gates Foundation, Nigeria Country Office, Global Development Division, Abuja, Nigeria

* wdevans@gwu.edu

## Abstract

### Background

In low- and middle-income countries (LMIC), group antenatal care (gANC) has the potential to reshape the traditional antenatal care (ANC) service delivery model, enhancing the care experience for both women and providers. ANC non-utilization rates are high in some parts of Nigeria, the evidence-based gANC program offers a structured yet flexible model that may increase maternal and child healthcare utilization which may in turn improve outcomes.

### Methods

This is a longitudinal study of women who participated in the gANC program at healthcare facilities sampled in Kaduna and Kano states, Nigeria. We plan to follow this longitudinal cohort over 3 time points of 18 months and to use data on participation in the gANC program as an independent variable to predict program outcomes using multivariate analysis and propensity matching techniques. This is aimed at isolating the causal effect of the number of gANC meetings attended during the index pregnancy on the probability of delivering that pregnancy in a health facility. To achieve that objective, we used inverse-probability weighting in addition to adjusted multivariate logistic regression models.

### Results

Overall, we found that there was high retention at follow-up, and generally high attendance at follow-up, with higher attendance based on other variables such as prior pregnancy and participants who were employed. We had low participants' attrition well within power analysis assumptions for the study. In terms of causal inference,

**Data availability statement:** All relevant data and code are within the paper and its Supporting Information files.

**Funding:** Gates Foundation grant INV-043363 The funders had no role in study design, data collection and analysis, decision to publish, or preparation of the manuscript.

**Competing interests:** The authors have declared that no competing interests exist.

there is a strong positive relationship between gANC session attendance and facility delivery, with women who attended five or more gANC sessions being approximately twice as likely to deliver in a health facility as those who attended none or one. Adjustment for a set of socio-demographic and prior pregnancy- and delivery-related variables via inverse probability weighting showed that a positive effect on facility delivery persists, especially at the highest levels of gANC session attendance.

## Discussion

Results of this study tend to confirm the main findings from previous studies of gANC, which found that higher levels of participation resulted in higher facility delivery. This study found that it was both scalable in a diverse set of urban and rural healthcare facilities and engagement with the program among eligible women was high. Future research in the current longitudinal study will evaluate long-term effects of gANC on maternal and child health outcomes such as family planning and vaccination.

## Conclusion

The gANC program at scale in Nigeria produced high levels of participation, and resulted in increased facility birthing utilization that were consistent with previous research. Future research should examine how to optimize program impact and enhance sustainability.

---

## Introduction

In low- and middle-income countries (LMIC), group antenatal care (gANC) has the potential to reshape the traditional antenatal care (ANC) service delivery model, enhancing the care experience for both women and providers [1]. In Nigeria, some regions in the north of the country show ANC non-utilization rates of over 50% [2]. Research is needed to understand how to overcome barriers to, and increase utilization of ANC in LMIC, especially among underserved regions of countries such as Nigeria.

Financial, access, and other barriers limit participation in antenatal care [3]. Moreover, there is no strong evidence that standard antenatal education improves birth or parenthood outcomes. In contrast, gANC offers a structured yet flexible model that integrates key components, including continuity of midwifery care, social support, and inquiry-based learning, each of which may have distinct mechanisms of impact. A systematic realist review identified six interconnected mechanisms through which gANC can enhance maternal care experiences: social support, peer learning, active health participation, health education, and satisfaction or engagement with care [4]. Additionally, shared health activities and women-led, group-based discussions have been shown to foster more equal and trusting relationships between women and midwives [5,6].

One critical area where gANC may provide significant benefits is in improving maternal knowledge and health behaviors. Studies indicate that maternal awareness of newborn danger signs and optimal newborn care practices is generally low. However, recognizing neonatal danger signs after birth strongly predicts care-seeking behavior for newborn illnesses. A study assessing the impact of gANC found that women in the intervention group had significantly higher knowledge of newborn danger signs and healthy newborn practices over time compared to those receiving individual ANC (iANC) [7].

Birth preparedness and complication readiness (BPCR) is another crucial factor influencing maternal and neonatal outcomes [8]. Limited BPCR has been identified as a major contributor to persistently high maternal and neonatal morbidity and mortality in low- and middle-income countries. BPCR includes raising awareness of danger signs, improving problem recognition, reducing delays in seeking care, choosing a birth location and provider in advance, knowing the nearest skilled provider, obtaining basic safe birth supplies, and identifying someone to accompany the woman to the facility during labor. A study found that BPCR scores significantly increased in the gANC group compared to the iANC group, suggesting that this model enhances women's preparedness for childbirth and emergencies.

The feasibility and acceptability of gANC have been evaluated in various settings. In Asia, particularly in India, a study explored the potential implementation of gANC, incorporating a clinical care component [9]. Stakeholders' perceptions, assessed through qualitative research, indicated strong support for the model following participation in a mock session demonstration. Women in gANC were also more likely to take prenatal vitamins and iron, use contraception at two months postpartum, and report higher satisfaction with the information provided, their relationship with healthcare providers, and overall quality of care.

Evidence from Ghana further supports the benefits of gANC. A cluster-randomized controlled trial comparing gANC with standard care found that by 34 weeks of gestation, women in the intervention group had significantly greater knowledge of family planning (FP) methods and a stronger intention to use FP after childbirth [10]. FP uptake was also consistently higher in the intervention group at all post-birth time points, except at 11–14 months postpartum, where the control group had higher rates.

Beyond family planning, gANC may also contribute to improved maternal health outcomes by increasing the uptake of key malaria in pregnancy (MIP) interventions. Malaria in pregnancy accounts for approximately 20% of stillbirths in sub-Saharan Africa and causes 10,000 maternal deaths worldwide each year [11]. A study conducted in Nigeria and Kenya suggests that gANC may enhance the uptake of intermittent preventive treatment in pregnancy with sulfadoxine-pyrimethamine (IPTp-SP), likely due to improved ANC retention [12]. Women in the intervention group had significantly higher IPTp-SP uptake compared to those receiving I-ANC (Nigeria: 3.45 vs. 2.14, $p < 0.001$; Kenya: 3.81 vs. 2.72, $p < 0.001$).

Overall, gANC presents a promising alternative to conventional antenatal care by integrating education, peer support, and active health engagement. Its ability to improve maternal knowledge, preparedness for birth, postnatal care utilization, and uptake of key maternal health interventions highlights its potential to transform ANC service delivery, particularly in resource-limited settings.

The present study reports on a follow-up (midline) of a large-scale gANC implementation project in Kaduna and Kano states in northern Nigeria. These areas have been identified by the Nigerian government, and the Gates Foundation, as priorities for interventions to improve maternal and child health (MCH) and to increase ANC utilization and care. Challenges include high maternal mortality rates, low availability of skilled birth attendants, and low ANC utilization in both states, as documented in previous research [13].

This is the first large-scale, quasi-experimental, longitudinal evaluation of gANC in Nigeria. Midline study findings fill a gap in knowledge about real-world implementation and effectiveness in these settings. The first phase (i.e., the baseline) of the study was implemented in Kaduna and Kano states in Nigeria in March and April 2024 [14]. The second phase of the project, follow-up data collection from the longitudinal cohort of women recruited in the first phase, was implemented

in Kaduna and Kano in October and November 2024. This study reports on analysis of the follow-up midline data from the second phase of the project. A future study will report on final findings from the project during the endline evaluation.

The key research questions addressed in this second midline phase of the project addressed implementation and primary effectiveness of gANC in promoting healthcare facility utilization. For instance, we asked: 1) To what extent was the gANC model implemented in Kaduna and Kano, how successful was the study in following women recruited in the first phase? 2) To what extent did attending more gANC sessions make women in our sample more likely to deliver in a health facility.

## Methods

### Study design

The full study design has been described in a previous publication [14]. Briefly, this is a longitudinal study of women who participated in the gANC program at healthcare facilities sampled in Kaduna and Kano states, Nigeria. We plan to follow this longitudinal cohort over 3 time points of 18 months and to use data on participation in the gANC program as an independent variable to predict program outcomes using multivariate analysis and analytic matching techniques [15].

The sampling plan for our study has been described in detail elsewhere [14]. Briefly, we assume retention of 80% of participants at follow-up, meaning that we will be able to determine whether they delivered in a health facility. This is a conservative estimate based on the figures in the Grenier et al. (2019) cRCT [16]. Based on our power calculations, the required baseline sample is 1,000 per state. In order to ensure more than sufficient statistical power, and to account for any issues in terms of lower-than-expected retention at follow up, we increased this total amount to 1,200. Because we also want to conduct stratified analyses by state, we need to recruit 1,200 women per state, for a total sample of 2,400 women. The actual recruited and retained sample is described below.

This paper focuses on changes in key variables among the longitudinal sample from baseline to midline follow-up data. The main outcome of interest is delivery of the index pregnancy measured at baseline in a healthcare facility among women who had some level of participation in the gANC program. The third and final wave (i.e., endline) of the longitudinal study will examine measures of utilization of and satisfaction with care for gANC services over the course of the pregnancy, as well as perinatal outcomes including facility delivery, completion of postnatal health checks, postpartum family planning use, and uptake of recommended child immunizations.

### Intervention

The gANC intervention has been described in detail in previous publications [14,16]. Briefly, the intervention consists of which includes group social support and ANC counseling, which are theorized to increase satisfaction with and intention to utilize healthcare facilities for prenatal care and birthing [16]. This study is based on a Theory of Change (ToC) that has been documented elsewhere [14]. The ToC is underpinned by the Diffusion of Innovation theory that explores how gANC activities are adopted and becomes a standard of care in northern Nigeria [17]. This model provides a basis for the conceptual framework of the program, which posits that ANC utilization can be influenced to promote the adoption of change, which in this case is the gANC intervention [18]. Observing the implementation of gANC in real life can provide important insight into facilitators and barriers that should be considered in planning for the scaling of the intervention in the States [19].

### Data collection

Baseline data collection for this study was conducted in Kaduna and Kano States from 25 March to 8 April of 2024. The follow-up midline data collection was conducted in these same locations from 27 October to 20 November of 2024.

A 154-item Questionnaire was used for the quantitative midline survey. The survey tool was pilot-tested prior to implementation. Data collection was conducted during 15 days of fieldwork in October and November 2024. Eligible women

(age 15–49 years at 12–20 weeks GA) who were recruited from randomly selected sample of facilities in each state at baseline were recontacted at follow-up. In each state, the quantitative survey was composed of three broad teams of interviewers and supervisors, one team per senatorial district. In each senatorial district, 10 interviewers and two supervisors conducted fieldwork. In-depth training on enumeration, interview protocol, and follow-up procedures were conducted before each survey wave. All interviewers were female to match the participants.

## Measures

The focal independent variable is number of gANC meetings attended during the index pregnancy. This was assessed by a series of questions in the follow-up questionnaire. The first question read, "Have you ever attended any group ANC meetings?" Those who answered affirmatively were then asked about their attendance at each of five gANC sessions: (1) preventing problems during pregnancy; (2) recognizing problems during pregnancy and joint planning on topics for male engagement; (3) birth preparedness, complication readiness, and health timing and spacing of pregnancies; (4) review of individual plans and family planning intentions; and (5) what to expect in labor and birth, recognizing postpartum problems, exclusive breastfeeding, and final birth planning. Using responses to these questions we computed the total number of gANC meetings attended by each participant, ranging from zero to five.

The focal dependent variable is delivery of the index pregnancy at a health facility (as opposed to at home or elsewhere). This was assessed by a series of two questions in the follow-up questionnaire. First, participants were asked, "Did you identify a facility for the delivery of your pregnancy?" Those who answered affirmatively were then asked, "Did you deliver at the facility you identified?" Based on responses to these questions we classified participants as having delivered the index pregnancy in a health facility, or not.

Our analysis also makes use of twelve covariates that were assessed at baseline. Five of these were basic sociodemographic variables: state of residence, urban versus rural residence, age, education, and employment. The remaining seven focused on aspects of prior pregnancy, delivery, and postpartum experiences. All participants were asked whether they had ever had a previous live birth, and how many living children they had. Those who had at least one prior live birth were asked whether they experienced problems in their most recent prior pregnancy, whether they delivered that pregnancy in a health facility, whether the delivery included a skilled birth attendant, whether they received a postnatal health check within one week following the delivery, and whether they received any postnatal health checks following that delivery.

## Data analysis

Our analysis of the data focused on participants who were located and reinterviewed at follow-up and whose index pregnancies had ended in a live birth, and proceeded as follows. After merging the baseline and follow-up data, we first examined attrition and retention. We obtained the number and percentage of baseline participants who were lost to follow-up and tabulated the reasons for attrition as reported by data collection personnel. We then examined patterns of attrition by cross-tabulating baseline socio-demographic variables (age, education, employment, urban vs rural residence, and parity); and, among those who had given birth at least once prior to the index pregnancy, cross-tabulating variables characterizing participants' most recent pregnancy, birth, and postpartum experiences. We used chi-squared tests of independence to determine whether levels of attrition were statistically significantly different across levels of those baseline sociodemographic and prior pregnancy variables. Because attrition was generally low and not strongly associated with baseline covariates (see Supplementary S1 Table in S1 File), all subsequent analyses focused on complete cases without multiple imputation or other method for addressing missing data.

Next, we examined participation in gANC meetings (our focal independent variable) among participants whose index pregnancy had ended in a live birth prior to the follow-up interview. We obtained the count and percent of participants

who attended each of five gANC meetings; computed the total number of gANC meetings each participant had attended; obtained counts and percents of the number of participants attending zero, one, two, three, four, and five gANC sessions; and obtained the mean number of gANC meetings attended. We then examined patterns of gANC meeting attendance by comparing the mean number of sessions attended across categories of baseline socio-demographic variables (age, education, employment, urban vs rural residence, and parity); and, among those who had given birth at least once prior to the index pregnancy, by comparing the mean number of gANC sessions attended across values of variables characterizing participants' most recent pregnancy, birth, and postpartum experiences. We used one-way analysis of variance to determine whether the mean number of sessions attended was statistically significantly different across levels of these baseline sociodemographic and prior pregnancy variables.

We then turned our attention to the primary endpoint: facility delivery. Among participants whose index pregnancy had ended in a live birth prior to the follow-up interview, we obtained the count and percent who reported delivering the index pregnancy in a health facility. This was measured both through our survey, and validation of delivery records at the facilities in Kaduna and Kano. We then examined whether the proportion of participants whose delivery occurred in a health facility varied in relation to baseline socio-demographic variables (age, education, employment, urban vs rural residence, and parity); and, among those who had given birth at least once prior to the index pregnancy, by values of variables characterizing participants' most recent prior pregnancy, birth, and postpartum experiences. These analyses were conducted via cross-tabulations, and we use chi-squared tests of independence to determine whether variations in the prevalence of facility delivery across socio-demographic and prior pregnancy variables were statistically significant.

Finally, we carried out a series of analyses to answer our main research question: Whether attending more gANC sessions makes women more likely to deliver in a health facility. We first examine whether the proportion delivering in a health facility varies in relation to number of gANC meetings attended by cross-tabulating those two variables and carrying out a chi-squared test of independence. We then run two logistic regression models, both with facility delivery as the dependent variable and number of gANC sessions attended as the focal independent variable. As a measure of model building, our first model had no control variables; the second model controlled for baseline sociodemographic variables. The purpose of this analysis is to determine whether the association between gANC meeting attendance and facility delivery is independent of potential confounding variables.

The objective of this study is to isolate the causal effect of number of gANC meetings attended during the index pregnancy on the probability of delivering that pregnancy in a health facility. To achieve that objective, we used a further analytic approach called inverse-probability weighting [20,21]. This approach is similar to propensity score methods in that a statistical model is used to collapse all of the information in the entire set of covariates relevant to treatment assignment into a single number. Inverse-probability weighting uses the inverse of the propensity score as a weight for treated participants, and the inverse of the complement of the propensity score as a weight for untreated participants. This approach generalizes to independent variables, like number of gANC meetings attended, that take more than two values. A multinomial logistic regression model is used to relate the set of covariates to the number of gANC visits (see Supplementary Table S2 in S1 File), and from this model we obtain for each participant the predicted probability the number of gANC sessions that she attended. Unweighted and weighted balance stastics for covariates in the inverse-probability weighting model are shown in Supplementary Table S3 in S1 File. The inverse of this predicted probability is then used as a weight in a model (in this case, logistic regression) relating the gANC meetings attended to the outcome of facility delivery. As a point of comparison, we also ran a more conventional regression-adjusted approach via a logistic regression model with facility delivery as the dependent variable, and number of gANC meetings attended and all twelve covariates as the independent variables.

Prior to running the inverse-probability weighted regression analysis, we first characterize the sample in terms of the distribution of all baseline covariates, number of gANC meetings attended, and facility delivery using simple frequencies. We next examine the associations between each of the twelve covariates and number of gANC meetings attended

by obtaining the mean and standard deviation of number of sessions attended within each level of each covariate, and by running linear regressions in which each covariate predicts number of sessions attended. This approach is justified in spite of the skewness of the gANC meeting attendance variable because inference in linear regression is robust to moderate violations of normality assumptions, and because this facilitates the incorporation of clustering of participants within local government authorities as described below. Following this, we examine the associations between the twelve covariates and facility delivery. In the linear and logistic regression models described here, we use cluster rather than conventional standard errors, with local government authority as the clustering variable, in order to obtain p-values that accurately reflect the study's complex sampling design. We use the same approach of clustered standard errors in our inverse-probability weighting analyses. All data management and analyses were carried out in Stata 18.0 [22]. For the inverse-probability weighting analysis we use Stata's *teffects ipwra* command.

## Research ethics

Ethical approval was obtained from the Nigeria's National Health Research Ethics Committee (NHREC). An ethical waiver was obtained from the Institutional Research Board (IRB) of The George Washington University. Only participants who gave informed consent were enrolled into the study. Informed consent was obtained verbally based on an IRB-approved statement read by a trained interviewer and witnessed by a second interviewer and recorded. Privacy and confidentiality of data were maintained throughout data collection, storage, and analysis. As this study was not a clinical trial, no trial registration was obtained.

## Results

Of the original sample of 2,469 pregnant women, 2,151 (87.1%) were located and reinterviewed at follow-up, which is a higher retention rate that required in our sampling plan. Of the 318 who were lost to follow-up, most (117, 71.4%) had moved away from the study area, while some (21, 6.6%) had died, and the remainder (70, 22.0%) refused to participate or were otherwise unavailable at follow-up throughout the data collection period. Also excluded from the analysis are 199 who had lost their pregnancies, and 12 who were still pregnant with the index pregnancy at the time of follow-up.

Table 1 shows the composition of the analytic sample of n = 1,952 in terms of socio-demographic background and aspects of prior pregnancy experiences. The sample is close to evenly distributed between the two study states of Kaduna and Kano, and is heterogeneous with respect to age, education, employment status, and number of living children. Participants were more likely to reside in rural areas than in urban areas, and just over three-quarters had given birth before. With respect to prior pregnancy experiences, about one-quarter had experienced problems in their most recent prior pregnancy, and approximately equal numbers had delivered their most recent pregnancy in a health facility as opposed to at home. Most prior deliveries involved the presence of a skilled birth attendant, although a substantial minority occurred without one. Most reported having had a postnatal health check within one week of their most recent delivery, and an even larger fraction had at least one postnatal care appointment following that delivery.

Table 2 shows the distribution of the focal independent and dependent variables: number of gANC meetings attended during the index pregnancy, and whether the index pregnancy was delivered in a health facility or not, respectively. In terms of number of gANC sessions attended, the distribution is skewed toward higher values, with five sessions being the modal value, followed by four sessions, and so on. A non-trivial proportion of just under 10% reported attending no gANC sessions at all during the index pregnancy. Just under half of the participants reported delivering the index pregnancy in a health facility.

Associations between the covariates and number of gANC meetings attended during the index pregnancy are shown in Table 3. The average number of meetings attended by study participants was 3.23, and this varied significantly across the states, with the average participant in Kaduna attending nearly an entire additional meeting than the average participant in Kano (p = 0.012). None of the differences across levels of the other variables were statistically significant, but some

**Table 1. Distribution of Sociodemographic and Prior Pregnancy- and Delivery-Related Variables among Participants in the Analytic Sample (n = 1952).**

|  | N | Percent |
|---|---|---|
| State |  |  |
| Kaduna | 906 | 46.4 |
| Kano | 1,046 | 53.6 |
| Age Group |  |  |
| 15-19 | 259 | 13.3 |
| 20-24 | 702 | 36.0 |
| 25-29 | 498 | 25.5 |
| 30-34 | 296 | 15.2 |
| 35+ | 197 | 10.1 |
| Education |  |  |
| Never attended formal school | 486 | 24.9 |
| Primary | 412 | 21.1 |
| Secondary | 717 | 36.7 |
| Higher | 198 | 10.1 |
| Qur'anic/Islamiyya | 133 | 6.8 |
| Other | 6 | 0.3 |
| Employment Status |  |  |
| Unemployed | 818 | 41.9 |
| Employed | 101 | 5.2 |
| Own a business | 1,033 | 52.9 |
| Residence |  |  |
| Urban | 721 | 36.9 |
| Rural | 1,231 | 63.1 |
| Ever Given Birth Before |  |  |
| Yes | 1,478 | 75.7 |
| No | 474 | 24.3 |
| Number of Living Children |  |  |
| 0 | 506 | 25.9 |
| 1 | 363 | 18.6 |
| 2 | 301 | 15.4 |
| 3 | 270 | 13.8 |
| 4 | 216 | 11.1 |
| 5+ | 296 | 15.2 |
| Experienced Problems During Prior Pregnancy/Delivery |  |  |
| No prior pregnancy/delivery | 404 | 20.7 |
| Yes | 516 | 26.4 |
| No | 1,032 | 52.9 |
| Place of Delivery for Last Pregnancy |  |  |
| No prior pregnancy/delivery | 404 | 20.7 |
| Home | 773 | 39.6 |
| Facility | 736 | 37.7 |
| Other | 39 | 2.0 |
| Skilled Birth Attended at Last Delivery |  |  |
| No prior pregnancy/delivery | 404 | 20.7 |
| Yes | 990 | 50.7 |
| No or Don't know | 558 | 28.6 |

*(Continued)*

**Table 1.** (Continued)

| | N | Percent |
|---|---|---|
| Postnatal Health Check within One Week of Last Delivery | | |
| No prior pregnancy/delivery | 404 | 20.7 |
| Yes | 1,273 | 65.2 |
| No or Don't know | 275 | 14.1 |
| Any Postnatal Care Appointments after Last Delivery | | |
| No prior pregnancy/delivery | 404 | 20.7 |
| Yes | 1,326 | 67.9 |
| No or Don't know | 222 | 11.4 |

**Table 2. Distribution of gANC Meeting Attendance and Facility Delivery (n = 1952).**

| Number of gANC Meetings Attended during Index Pregnancy | n | % |
|---|---|---|
| 0 | 180 | 9.2 |
| 1 | 168 | 8.6 |
| 2 | 254 | 13.0 |
| 3 | 367 | 18.8 |
| 4 | 395 | 20.2 |
| 5 | 588 | 30.1 |
| Index Pregnancy Delivered in a Health Facility | | |
| No | 1,010 | 51.7 |
| Yes | 942 | 48.3 |

were substantial in magnitude and came close to achieving statistical significance. Among those who had at least one prior delivery, for example, those who had their most recent delivery at a health facility attended an average of 3.45 gANC meetings during the index pregnancy, while those who delivered at home attended an average of 3.00. Likewise, those who reported attending a postnatal health check within one week of their prior delivery attended an average of 3.27 gANC meetings during the current pregnancy, while those who did not attended an average of just 2.94. The participants who reported being employed at baseline attended an average of 3.77 gANC meetings during the index pregnancy, compared to 3.06 among those who were not employed and an intermediate average of 3.30 among those employed in a self- or family-owned business. Together, these findings suggest that these covariates influence the number of gANC meetings attended to a modest degree.

Table 4 shows associations between the covariates and facility delivery. In general, these associations are more substantial than those between the covariates and number of gANC sessions attended, and almost all of the associations are statistically significant. For example, facility delivery was almost twice as common in Kaduna (64.0%) as in Kano (34.6%), and was positively associated with higher education, urban residence, and employment status. Facility delivery was more common among those who had never given birth before; and was negatively associated with the number of living children. Facility delivery was also positively associated with several indicators of health care utilization during prior pregnancy, including facility delivery during that pregnancy, the presence of a skilled birth attendant, and receipt of post-natal care.

The unadjusted and adjusted associations between number of gANC meetings attended during the index pregnancy and facility delivery are shown in Table 5. The percentage delivering in a health facility increases consistently from as low as 28.3% among those who attended no gANC meetings, to as high as 61.4% among those who attended five or more meetings. The statistical significance of these unadjusted differences is shown in the "Unadjusted" panel. While the

**Table 3. Associations between Background Variables and Number of Meetings Attended (n = 1952).**

| | Mean | (SD) | p-value |
|---|---|---|---|
| State | | | 0.012 |
| Kaduna | 3.73 | (1.37) | |
| Kano | 2.79 | (1.71) | |
| Age Group | | | 0.390 |
| 15-19 | 3.05 | (1.70) | |
| 20-24 | 3.28 | (1.61) | |
| 25-29 | 3.33 | (1.62) | |
| 30-34 | 3.11 | (1.68) | |
| 35+ | 3.18 | (1.59) | |
| Education | | | 0.225 |
| Never attended formal school | 2.98 | (1.67) | |
| Primary | 3.18 | (1.64) | |
| Secondary | 3.32 | (1.61) | |
| Higher | 3.68 | (1.50) | |
| Qur'anic/Islamiyya | 3.06 | (1.63) | |
| Other | 3.67 | (1.21) | |
| Employment | | | 0.109 |
| Unemployed | 3.06 | (1.69) | |
| Employed | 3.77 | (1.48) | |
| Own a business | 3.30 | (1.58) | |
| Residence | | | 0.612 |
| Urban | 3.36 | (1.57) | |
| Rural | 3.15 | (1.66) | |
| Ever Given Birth Before | | | 0.936 |
| Yes | 3.22 | (1.63) | |
| No | 3.23 | (1.64) | |
| Number of Living Children | | | 0.372 |
| 0 | 3.23 | (1.65) | |
| 1 | 3.25 | (1.63) | |
| 2 | 3.31 | (1.70) | |
| 3 | 3.34 | (1.59) | |
| 4 | 3.13 | (1.65) | |
| 5+ | 3.07 | (1.55) | |
| Experienced Problems During Prior Pregnancy/Delivery | | | 0.149 |
| No prior pregnancy/delivery | 3.29 | (1.62) | |
| Yes | 3.05 | (1.70) | |
| No | 3.28 | (1.59) | |
| Place of Delivery for Last Pregnancy | | | 0.091 |
| No prior pregnancy/delivery | 3.29 | (1.62) | |
| Home | 3.00 | (1.68) | |
| Facility | 3.45 | (1.55) | |
| Other | 2.74 | (1.71) | |
| Skilled Birth Attended at Last Delivery | | | 0.234 |
| No prior pregnancy/delivery | 3.29 | (1.62) | |
| Yes | 3.33 | (1.58) | |
| No or Don't know | 2.99 | (1.71) | |

*(Continued)*

**Table 3.** (Continued)

| | Mean | (SD) | p-value |
|---|---|---|---|
| Postnatal Health Check within One Week of Last Delivery | | | 0.107 |
| No prior pregnancy/delivery | 3.29 | (1.62) | |
| Yes | 3.27 | (1.62) | |
| No or Don't know | 2.94 | (1.67) | |
| Any Postnatal Care Appointments after Last Delivery | | | 0.193 |
| No prior pregnancy/delivery | 3.29 | (1.62) | |
| Yes | 3.27 | (1.61) | |
| No or Don't know | 2.85 | (1.74) | |
| Total | 3.23 | (1.63) | |

prevalence of facility delivery was not statistically distinguishable among those who attended no meetings or those who attended two meetings compared to those who attended only one, the odds of facility delivery were substantially higher among those attending three, four, or five or more meetings. That basic pattern remains when conventional regression adjustment is used to address confounding, as shown in the "Regression Adjusted" panel of Table 5. While the adjusted odds ratios for three, four, and five meetings are somewhat closer to the null than in the unadjusted model, all three remain positive and statistically significant. Much more substantial attenuation in the association between gANC meeting attendance and facility delivery when inverse probability weighting is used to address confounding. In that analysis, the effect of attending three as opposed to one meeting is no longer statistically significant. The effects of attending four or five or more gANC meetings remain statistically significant, but are much smaller in magnitude.

## Discussion

In this study, we examined: 1) implementation of gANC intervention and longitudinal follow-up of women who participated in gANC, and 2) whether attending a larger number of gANC sessions increased the likelihood of facility delivery among pregnant women in Kaduna and Kano states in Nigeria.

In terms of the first study aim, we found that there was high retention at follow-up (87.1%), and generally high attendance at follow-up (3.23 gANC meetings), with higher attendance based on other variables such as prior pregnancy and participants who were employed. This is a significant achievement in a study in low resource settings such as Kaduna and Kano. In addition, participation rates skewed to the high end of the scale of possible meetings attended, with the modal participant rate estimated at 5 gANC meetings, and next highest being 4 meetings. Overall, gANC implementation proceeded as planned, and consistent with prior studies, [15] and our attrition rate was lower than projected for power analysis.

In terms of the second aim, there is a strong positive relationship between gANC session attendance and facility delivery, with women who attended give or more gANC sessions being approximately twice as likely to deliver in a health facility as those who attended none or one. Adjustment for a set of socio-demographic and prior pregnancy- and delivery-related variables via inverse probability weighting, however, substantially attenuates that association. Evidence of a positive effect on facility delivery persists, especially at the highest levels of gANC session attendance. It is evident, however, that simple or even conventional regression-adjusted estimates substantially overstate the impact of gANC meeting attendance on facility delivery.

Overall, results of this study tend to confirm the main findings from previous studies of gANC, which found that higher levels of participation resulted in higher facility delivery [16]. This study examined implementation of gANC at scale in 2 large Nigerian states, and found that it is both scalable in a diverse set of urban and rural healthcare facilities, and engagement (in terms of meeting attendance) with the program among eligible women was high. The program is a promising strategy to promote maternal and child health in LMICs [23].

**Table 4. Associations between Background Variables and Percent Delivering in a Health Facility (n = 1952).**

| | Percent | p-value |
|---|---|---|
| State | | 0.001 |
| Kaduna | 64.0 | |
| Kano | 34.6 | |
| Age Group | | 0.252 |
| 15-19 | 47.1 | |
| 20-24 | 52.1 | |
| 25-29 | 45.8 | |
| 30-34 | 46.0 | |
| 35+ | 45.7 | |
| Education | | 0.000 |
| Never attended formal school | 30.9 | |
| Primary | 44.7 | |
| Secondary | 58.0 | |
| Higher | 68.7 | |
| Qur'anic/Islamiyya | 39.9 | |
| Other | 50.0 | |
| Employment | | 0.055 |
| Unemployed | 46.5 | |
| Employed | 60.4 | |
| Own a business | 48.5 | |
| Residence | | 0.000 |
| Urban | 64.5 | |
| Rural | 38.8 | |
| Ever Given Birth Before | | 0.009 |
| Yes | 45.4 | |
| No | 57.2 | |
| Number of Living Children | | 0.003 |
| 0 | 56.3 | |
| 1 | 51.5 | |
| 2 | 44.2 | |
| 3 | 49.6 | |
| 4 | 41.7 | |
| 5+ | 38.2 | |
| Experienced Problems During Prior Pregnancy/Delivery | | 0.004 |
| No prior pregnancy/delivery | 57.2 | |
| Yes | 45.0 | |
| No | 46.4 | |
| Place of Delivery for Last Pregnancy | | 0.000 |
| No prior pregnancy/delivery | 57.2 | |
| Home | 35.2 | |
| Facility | 56.3 | |
| Other | 64.1 | |
| Skilled Birth Attended at Last Delivery | | 0.000 |
| No prior pregnancy/delivery | 57.2 | |
| Yes | 49.9 | |
| No or Don't know | 38.9 | |

*(Continued)*

PLOS One | https://doi.org/10.1371/journal.pone.0333383   October 8, 2025                                    12 / 15

**Table 4.** (Continued)

| | Percent | p-value |
|---|---|---|
| Postnatal Health Check within One Week of Last Delivery | | 0.000 |
| No prior pregnancy/delivery | 57.2 | |
| Yes | 47.0 | |
| No or Don't know | 41.1 | |
| Any Postnatal Care Appointments after Last Delivery | | 0.000 |
| No prior pregnancy/delivery | 57.2 | |
| Yes | 46.7 | |
| No or Don't know | 41.4 | |

**Table 5. Unadjusted and Adjusted Associations between Number of gANC Meetings Attended and Delivery in a Health Facility (n = 1952).**

| Number of gANC Meetings Attended | Percent Delivering in a Health Facility | Unadjusted | | Regression Adjusted | | IPW Adjusted | |
|---|---|---|---|---|---|---|---|
| | | OR | (95% C.I.) | AOR | (95% C.I.) | AOR | (95% C.I.) |
| 0 | 28.3 | 0.79 | (0.47-1.34) | 1.03 | (0.62-1.72) | 1.11 | (0.89-1.33) |
| 1 | 33.3 | 1.00 | (reference) | 1.00 | (reference) | 1.00 | (reference) |
| 2 | 37.8 | 1.22 | (0.75-1.98) | 1.29 | (0.77-2.28) | 1.06 | (0.93-1.18) |
| 3 | 47.7 | 1.82 | (1.17-2.83)** | 1.73 | (1.12-2.68)* | 1.13 | (0.99-1.26) |
| 4 | 51.4 | 2.11 | (1.34-3.35)** | 2.01 | (1.33-3.02)** | 1.16 | (1.03-1.28)** |
| 5 | 61.4 | 3.18 | (1.60-6.33)** | 2.73 | (1.45-5.14)** | 1.24 | (1.04-1.44)** |

## Limitations

Two main limitations of this study should be noted. First, both the focal independent variable of number of gANC meetings attended, as well as the dependent variable of facility delivery of the index pregnancy, were assessed via participant self-report. This could lead to upward or downward bias in our observed associations between gANC meeting attendance and facility delivery. Both gANC meeting attendance and facility delivery may be subject to some degree of social desirability bias, and some participants may be more inclined than others to provide socially desirable responses.

The second main limitation is the possibility of omitted variable bias. While we had a substantial set of covariates that included both sociodemographic background variables as well as characteristics of participants' previous pregnancy, delivery, and postpartum experiences, other confounders could have been omitted from this set. This concern may be mitigated to some extent by the associations between included covariates and omitted confounders. One possible confounder that was not included in our analysis, for example, was the distance between each participant's dwelling and the health facility, or more generally the difficulty involved in traveling from the dwelling to the facility. Such omitted variable bias may have affected results of the inverse-probability weighted analyses.

## Future research

This is the second in a series of three longitudinal studies on the gANC program in Nigeria. Future research in this study at endline will examine additional, long-term outcomes of program participation, including satisfaction with and utilization

of facility-based healthcare, family planning utilization, contraception, and infant and child health services such as vaccination. Beyond the current study, research should examine methods to optimize gANC, such as isolating specific services that are most engaging, utilized, and efficacious in promoting maternal and child health behaviors [24]. Implementation studies that examine specific strategies for delivering services should be conducted to identify specific, effective program components. One possible approach would be to examine if promotion of gANC through channels such as digital media, such as social media or patient-provider apps, would increase program participation and intended outcomes discussed earlier [25].

## Conclusion

Implementation of the gANC program at scale in Nigeria demonstrated that the program is effective at scale. Findings were consistent with previous research. The program is a promising approach to promoting maternal and child healthcare in LMICs. Future studies should examine long-term effects of the program on participants and ways to optimize gANC implementation.

## Supporting information

**S1 File. Supplementary Tables S1-S3.**
(DOCX)

## Author contributions

**Conceptualization:** William  Douglas Evans, Samson Babatunde Adebayo, Sani Ali Gar, Masduk Abdulkarim.

**Data curation:** Jeffrey Bartlett Bingenheimer, Taiseer Zaman, Sani Ali Gar.

**Formal analysis:** William Douglas Evans, Jeffrey Bartlett Bingenheimer.

**Funding acquisition:** William  Douglas Evans, Samson Babatunde Adebayo.

**Investigation:** William  Douglas Evans, Chinwe Lucia Ochu, Samson Babatunde Adebayo, Fasiku Adekunle David.

**Methodology:** William  Douglas Evans, Jeffrey Bartlett Bingenheimer.

**Project administration:** William  Douglas Evans, Fasiku Adekunle David, Sani Ali Gar.

**Supervision:** William  Douglas Evans.

**Visualization:** Taiseer Zaman.

**Writing – original draft:** William  Douglas Evans.

**Writing – review & editing:** Jeffrey Bartlett Bingenheimer, Taiseer Zaman, Chinwe Lucia Ochu, Samson Babatunde Adebayo.

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
