## [Decision Letter · Decision Letter 0]

11 Aug 2025

 Thank you for submitting your manuscript to PLOS ONE. After careful consideration, we feel that it has merit but does not fully meet PLOS ONE’s publication criteria as it currently stands. Therefore, we invite you to submit a revised version of the manuscript that addresses the points raised during the review process. In addition to the points raised by the reviewers, I would like to bring your attention to the statistical analysis used for Table 3. Did you verify whether your data (Number of Meetings Attended) met the assumptions required for using parametric tests, specifically the assumption of normality, before applying the ANOVA? Additionally, I am curious why you chose to present the association using means and standard deviations, rather than treating the "Number of Meetings Attended" as a categorical variable (nominal or ordinal) and using the Chi-square test to obtain your p-values, especially since you ultimately treated this variable as categorical in Table 5.

We look forward to receiving your revised manuscript.

Kind regards,

Ayodeji Babatunde Oginni

Academic Editor

PLOS ONE

Journal Requirements:

“Gates Foundation grant INV-043363”

5. We note that you have indicated that there are restrictions to data sharing for this study. PLOS only allows data to be available upon request if there are legal or ethical restrictions on sharing data publicly. For more information on unacceptable data access restrictions, please see http://journals.plos.org/plosone/s/data-availability#loc-unacceptable-data-access-restrictions .  

6. Please include captions for your Supporting Information files at the end of your manuscript, and update any in-text citations to match accordingly. Please see our Supporting Information guidelines for more information: http://journals.plos.org/plosone/s/supporting-information .

Reviewers' comments:

Reviewer's Responses to Questions

**Comments to the Author**

1. Is the manuscript technically sound, and do the data support the conclusions?

Reviewer #1: Yes

Reviewer #2: No

Reviewer #3: Yes

Reviewer #4: Yes

2. Has the statistical analysis been performed appropriately and rigorously?

Reviewer #1: Yes

Reviewer #2: No

Reviewer #3: Yes

Reviewer #4: Yes

3. Have the authors made all data underlying the findings in their manuscript fully available?

Reviewer #1: No

Reviewer #2: No

Reviewer #3: Yes

Reviewer #4: No

4. Is the manuscript presented in an intelligible fashion and written in standard English?

Reviewer #1: Yes

Reviewer #2: No

Reviewer #3: Yes

Reviewer #4: Yes

Reviewer #1: comments

The introduction provides a comprehensive overview of the rationale for group antenatal care (gANC) in LMICs, especially Nigeria, and summarizes relevant evidence. However, it would benefit from a clearer articulation of what this specific study adds to the existing literature. For example, explicitly state how this is the first large-scale, quasi-experimental, longitudinal evaluation of gANC in Kaduna and Kano, and how its midline findings fill a gap in knowledge about real-world implementation and effectiveness in these settings

The introduction mentions high non-utilization rates of ANC in northern Nigeria but could more directly link this to the choice of Kaduna and Kano as study sites. Briefly highlight the distinct demographic, socioeconomic, and cultural factors in these states that make them both representative and challenging for ANC interventions

Use consistent terminology for group antenatal care (gANC or G-ANC) and individual antenatal care (I-ANC) throughout the introduction.

Provide more detail on the sampling strategy: How were healthcare facilities and participants selected? Was randomization used at any stage? Clarify inclusion and exclusion criteria, and discuss any potential for selection bias

The section describes the use of a 154-item questionnaire and fieldwork procedures, but could be strengthened by detailing: How interviewers were trained and monitored, How data quality was assured during fieldwork, Any steps taken to minimize interviewer bias or respondent misunderstanding

The methods should specify how missing data were addressed in the analysis (e.g., imputation, exclusion, sensitivity analyses), especially given the longitudinal design and potential for attrition

Describe how data for intervention (gANC) and comparison (I-ANC) groups were stored separately (e.g., encrypted databases with unique identifiers) to prevent contamination.

Emphasize the significance of the high retention (87.1%) and attendance rates in the context of known challenges with ANC utilization in northern Nigeria, highlighting how this supports the feasibility and acceptability of gANC in these settings.

Provide more detail on what aspects of the gANC implementation contributed to high attendance and retention (e.g., community mobilization, group dynamics, provider engagement), referencing qualitative findings if available.

Mention any challenges encountered during implementation and how they were addressed or could be mitigated in future scale-up.

Acknowledge the lack of randomization and potential for residual confounding despite inverse probability weighting.

Reviewer #2: Comments:

1. Lack of Clarity and Specificity in Background:

The background is overly generic and fails to clearly establish the research gap or the rationale for using gANC in the Nigerian context. It references high ANC non-utilization in "some parts of Nigeria" without citing specific statistics, regions, or studies.

The link between gANC and improved maternal outcomes is claimed without sufficient supporting literature. Assertions such as “gANC has the potential to reshape traditional ANC” are unsubstantiated and speculative.

No theoretical framework or conceptual model underpins the study’s hypothesis.

2. Weak and Ambiguous Methodology:

The study design lacks sufficient detail. It is described as "longitudinal," yet no timeline, follow-up intervals, or baseline characteristics are presented.

It is unclear how participants were recruited, how exposure to gANC was defined, or what specific inclusion/exclusion criteria were used.

The use of inverse probability weighting and propensity score matching is stated, but the manuscript does not describe the model diagnostics, variable selection process, or balance statistics. This undermines the validity of the causal claims.

There is no reporting of confidence intervals, p-values, or model fit statistics. The manuscript vaguely claims "positive association" without statistical evidence.

3. Results Are Descriptive and Lack Analytical Depth:

The results section reads like a narrative summary and fails to provide tabulated data, model outputs, or measures of uncertainty.

There is no stratification by relevant subgroups (e.g., rural vs. urban, parity, age).

Terms like "high attendance" and "low attrition" are vague. The reader is not told how many participants were recruited, followed up, or lost, nor how attrition may have biased the results.

4. Unsupported Causal Claims:

The manuscript repeatedly implies causality (“causal effect of gANC on facility delivery”) despite relying on observational data.

While statistical methods such as IPW are mentioned, the paper does not demonstrate their correct application or justify assumptions necessary for causal inference (e.g., no unmeasured confounding, positivity, consistency).

5. Poor Structure and Repetitive Content:

There is repetition of key points across sections (e.g., that attendance is associated with facility delivery is stated in background, results, discussion, and conclusion without added nuance).

The writing is vague, non-scientific, and sometimes speculative, e.g., "scalable in diverse urban and rural settings" is stated without scalability analysis or implementation details.

There is a lack of clear outcome definitions—e.g., how was facility delivery validated or measured?

6. No Contribution to Existing Literature:

The manuscript claims its results confirm prior findings but does not offer new insights, mechanisms, or innovations in delivery or evaluation of gANC.

There is no critical discussion of potential biases, limitations, or alternative explanations.

7. Ethical Considerations and Data Transparency Missing:

No mention of ethical approval, informed consent, or data protection procedures for participants in a longitudinal cohort study.

No indication of data availability, data sharing policies, or trial registration if applicable.

Reviewer #3: Dear Author,

Thank you for this study which is a good addition to antenatal care in Northern Nigeria.

I see this study as a simplified form of antenatal care with focus on improving maternal and child health outcomes.

Please attend to my concerns below:

TITLE: This title needs to be reviewed. I think the title should reflect the focus of the study more than the methodology. It should also be minimally encompassing and reflects the fraction of the study area as much as possible. suggested title would be like: Group Antenatal Care Intervention in Two Northern Nigerian states: A Quasi-experimental evaluation/study/assessment.

ABSTRACT: Good abstract. Clear and concise, well focused and informative. Thank you.

You may need to include the "keywords"

INTRODUCTION: Generally, this introduction meets the necessary requirement for the follow up phase of a longitudinal study. The social and scientific values of the study are well presented. Thank you.

Paragraph2: It is generic that standard/focused antenatal education improves maternal and fetal outcomes in any pregnancy. The level of influence/impact may be debatable and subject to other independent variables.

Please rephrase this statement.

MEASURES:

P1: You meant " presenting problems"?

P2: Measurement of this dependent variable would definitely reflect the successful impact of gANC to influence antenatal health seeking behaviors, however, i think measuring the pregnancy outcomes would be of equal clinical significance,

Consider adding this if data is available.

RESULTS:

P1: please use a uniform numerical representation: either x, y% or x (y%). Thanks

P1: Is this a pregnancy related death? Clarify if known.

TABLE2: This is very important in this study and also remarkable for a longitudinal study. Ability to shift this upwardly will determine the clinical and statistical impact of this study. Thank

FUTURE RESEARCH: I think this study needs to consider a mixed research method for the 3rd series of the study. I think a semi-structured or focus group interview of some of the participants will answer a lot of questions that will strengthen the goal of the study.

Reviewer #4: The authors have presented their work with sound argument and discuss their findings with rigour. However, because this is a follow up write up of a larger work it would have helped if they summarised earlier findings of the work to give context frothier findings in this work. This does not take away from the work presented but it could have improved on the presentation. This said, thank you for the opportunity to review this work.

The work is significant as maternal mortality remains a challenge in LMICs. Increasing coverage and improving affordability through gANC will go a long way if implemented adequately. I believe this work can be published as is.

**Do you want your identity to be public for this peer review?** For information about this choice, including consent withdrawal, please see our Privacy Policy

Reviewer #1: **Yes: ** Belayneh Jejaw Abate

Reviewer #2: No

Reviewer #3: **Yes: ** Adeloye Amoo Adeniji (MBBS; MMed; FCFP; FACRRM)

Reviewer #4: No

---

## [Author Response · Author response to Decision Letter 1]

3 Sep 2025

We have provided a file in the resubmission with our point-by-point responses to reviewer comments.

---

## [Editor Report · Decision Letter 1]

7 Sep 2025

Dear Dr. Evans,

**In terms of number of gANC sessions attended, the distribution is skewed toward higher values, with five sessions being the modal value, followed by four sessions, and so on.**I would like you to include a brief statement on this in the ?>**Data Analysis** section after this statement: "**We next examine the associations between each of the twelve covariates and number of gANC meetings attended by obtaining the mean and standard deviation of number of sessions attended within each level of each covariate** "    

We look forward to receiving your revised manuscript.

Kind regards,

Ayodeji Babatunde Oginni

Academic Editor

PLOS ONE
---

## [Author Response · Author response to Decision Letter 2]

11 Sep 2025

We have uploaded a response memo in the updated revision 2 files.

---

## [Editor Report · Decision Letter 2]

15 Sep 2025

Evaluation of a Group Antenatal Care Intervention in Two Northern Nigerian states: Quasi-experimental study

PONE-D-25-15448R2

Dear Dr. Evans,

We’re pleased to inform you that your manuscript has been judged scientifically suitable for publication and will be formally accepted for publication once it meets all outstanding technical requirements.

Kind regards,

Ayodeji Babatunde Oginni

Academic Editor

PLOS ONE
---

## [Editor Report · Acceptance letter]

PONE-D-25-15448R2

PLOS ONE

Dear Dr. Evans,

I'm pleased to inform you that your manuscript has been deemed suitable for publication in PLOS ONE. Congratulations! Your manuscript is now being handed over to our production team.

Kind regards,

on behalf of

Ayodeji Babatunde Oginni

Academic Editor

PLOS ONE